# Interventional Sum-Product Networks:
# Causal Inference with Tractable Probabilistic Models

**Matej Zečević**[1]     **Devendra Singh Dhami**[1]     **Athresh Karanam**[2]

**Sriraam Natarajan**[2]     **Kristian Kersting**[1,3]

[1]Computer Science Department, TU Darmstadt
[2]Computer Science Department, The University of Texas at Dallas
[3]Centre for Cognitive Science, TU Darmstadt, and Hessian Center for AI (hessian.AI)
`{matej.zecevic, devendra.dhami, kersting}@cs.tu-darmstadt.de`
`{athresh.karanam,sriraam.natarajan}@utdallas.edu`

## Abstract

While probabilistic models are an important tool for studying causality, doing so suffers from the intractability of inference. As a step towards tractable causal models, we consider the problem of learning interventional distributions using sum-product networks (SPNs) that are over-parameterized by gate functions, e.g., neural networks. Providing an arbitrarily intervened causal graph as input, effectively subsuming Pearl's *do*-operator, the gate function predicts the parameters of the SPN. The resulting *interventional* SPNs are motivated and illustrated by a structural causal model themed around personal health. Our empirical evaluation against competing methods from both generative and causal modelling demonstrates that interventional SPNs indeed are both expressive and causally adequate.

## 1   Introduction

Identifying causal relationships between variables in observational data is one of the fundamental and well-studied problem in machine learning. There have been several great strides in causality [Granger, 1969, Pearl, 2009, Bareinboim and Pearl, 2016] over the years specifically characterized by efforts that focused on reasoning about interventions [Hagmayer et al., 2007, Dasgupta et al., 2019] and counterfactuals [Morgan and Winship, 2015, Oberst and Sontag, 2019].

The notion of causality has long been explored in the realm of probabilistic models [Oaksford and Chater, 2017, Beckers and Halpern, 2019] with a special focus on graphical models, called causal Bayesian networks (CBNs) [Heckerman et al., 1995, Neapolitan, 2004, Pearl, 1995, Acharya et al., 2018]. CBNs have widely been applied to infer causal relationships in high-impact diverse applications such as disease progression [Koch et al., 2017], ecological risk assessment [Carriger and Barron, 2020] and more recently Covid-19 [Fenton et al., 2020, Feroze, 2020] to name a few. Although successful, classical CBN models are difficult to scale and also suffer from the problem of intractable inference. Recently, tractable probabilistic models such as probabilistic sentential decision diagrams [Kisa et al., 2014] and sum-product networks [Poon and Domingos, 2011] have emerged, which guarantee that conditional marginals can be computed in time linear in the size of the model. While weaving in the notion of interpretability, the computational view on probabilistic models allows one to exploit ideas from deep learning and can thus be very useful in modelling complex problems.

Recently, there has been an effort to take advantage of this tractability to reason for causality. Zhao et al. [2015] showed how to compile back and forth between sum-product networks (SPNs) and Bayesian networks (BNs). Although this opened up a whole range of possibilities for tractable

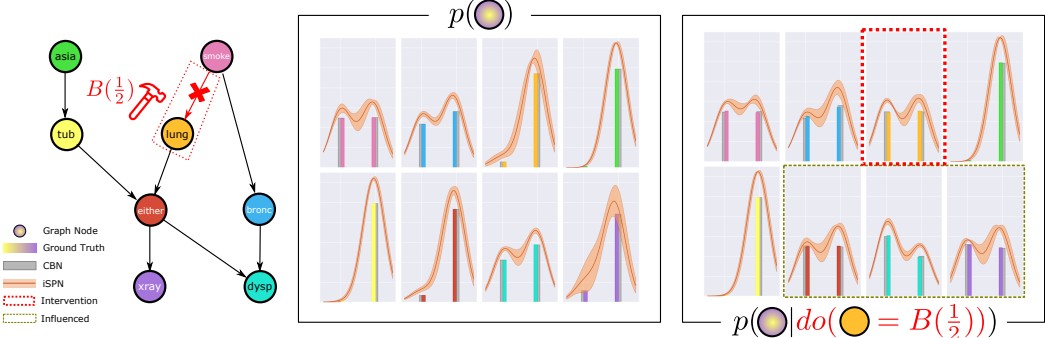

Figure 1: **Capturing interventional distributions using iSPN**. The interventional distributions for the ASIA data set using a causal Bayesian network (CBN, small-scale gold standard, gray bars) as well as an interventional SPN (iSPN) by intervening on *lung*. The iSPN is sensible to all the influences of the given intervention onto the system i.e., subsequent effects in the causal hierachy. (Best viewed in color.)

causal models, Papantonis and Belle [2020] argued that such conversion leads to degenerated BNs thereby rendering it ineffective for causal reasoning. For the considered compilation of SPNs to BNs, this is indeed the case since a bipartite graph between the hidden and observed variables loses the relationships between the actual variables. Thus, either a new compilation method for transforming between tractable and causal model or alternatively a method to condition the probabilistic models directly on the *do*-operator to obtain interventional distributions, $P(y \mid do(x))$, is being required.

Here, we consider the latter strategy and extend the idea of conditionally parameterizing SPNs [Shao et al., 2019] by conditioning on the *do*-operator while predicting the complete set of observed variables therefore capturing the effect of intervention(s). The resulting *interventional sum-product networks (iSPNs)* take advantage of both the expressivity, due to the neural network, and the tractability, due to the SPN in order to capture the interventional distributions faithfully. This shows that the dream of tractable causal models is not insurmountable, since iSPNs are causal. Pearl [2019] defined a three-level causal hierarchy that separates association (purely statistical level 1) from intervention (level 2) and counterfactuals (level 3), and argued that the latter two levels involve causal inference. iSPNs are a modelling scheme for arbitrary interventional distributions. They thus belong to level 2 and, in turn, are causal. So, while SPNs are not universal function approximators and the use of gate functions turns them into universal approximators, we go one step ahead and make the first effort towards introducing causality to SPNs without the need for compilation to Bayesian networks, as the functional approximator subsumes the *do*-operator. Fig. 1 shows an example of the effectiveness of iSPNs to capture interventional distributions on the ASIA data set. Our extensive experiments against strong baselines demonstrate that our method is able to capture ideal interventional distributions.

To summarize, we make the following contributions:

1. We introduce iSPNs, the first method that applies the idea of tractable probabilistic models to causality without the need for compilation and that accordingly generate interventional distributions w.r.t. the provided causal structure.

2. We formulate the inductive bias necessary for turning conditional SPN into iSPN while taking advantage of the neural network modelling capacities within the gating nodes for for modelling interventional distributions while capturing all influences within the given intervention (i.e., consequences propagated through the structural hierarchy).

3. We show that, by construction, iSPNs can identify any interventional distribution permitted by the underlying structural causal model due to the inducted bias on the interface modalities in junction with the universal function approximation of the underlying gated SPN.

We proceed as follows. We start by reviewing the basic concepts required and related work, namely the tractable model class of sum-product networks and key concepts from causality. Then we motivate using a newly curated causal model themed around personal health. Subsequently, we introduce iSPNs formally and prove that by construction they are capable of approximating any *do*-query (given corresponding data). Before concluding, we present our experimental evaluation in which we

challenge iSPN in its density estimation and causal inference capabilities against various baselines. We make our code publically available at: `https://github.com/zecevic-matej/iSPN`.

## 2 Background and Related Work

Let us briefly review the background on tractable probabilistic models and causal models used in subsequent sections for developing our new model class based on CSPNs that allow for identifying causal quantities i.e., interventional distributions.

**Notation.** We denote indices by lower-case letters, functions by the general form $g(\cdot)$, scalars or random variables interchangeably by upper-case letters, vectors, matrices and tensors with different boldface font $\mathbf{v}, \mathbf{V}, \mathsf{V}$ respectively, and probabilities of a set of random variables $\mathbf{X}$ as $p(\mathbf{X})$.

**Sum-Product Networks (SPNs).** Introduced by Poon and Domingos [2011], generalizing the notion of network polynomials based on indicator variables $\lambda_{X=x}(\mathbf{x}) \in [0, 1]$ for (finite-state) RVs $\mathbf{X}$ from [Darwiche, 2003], Sum-Product Networks (SPNs) represent a special type of probabilistic model that allows for a variety of exact and efficient inference routines. Generally, SPNs are considered as directed acyclic graphs (DAG) consisting of product, sum and leaf (or distribution) nodes whose structure and parameterization can be efficiently learned from data to allow for efficient modelling of joint probability distributions $p(\mathbf{X})$. Formally a SPN $\mathcal{S} = (G, \mathbf{w})$ consists of non-negative parameters $\mathbf{w}$ and a DAG $G = (V, E)$ with indicator variable $\boldsymbol{\lambda}$ leaf nodes and exclusively internal sum and product nodes given by,

$$\mathsf{S}(\boldsymbol{\lambda}) = \sum_{\mathsf{C} \in \text{ch}(\mathsf{S})} \mathbf{w}_{\mathsf{S},\mathsf{C}} \mathsf{C}(\boldsymbol{\lambda}) \quad \mathsf{P}(\boldsymbol{\lambda}) = \prod_{\mathsf{C} \in \text{ch}(\mathsf{S})} \mathsf{C}(\boldsymbol{\lambda}), \tag{1}$$

where the SPN output $\mathcal{S}$ is computed at the root node ($\mathcal{S}(\boldsymbol{\lambda}) = \mathcal{S}(\mathbf{x})$) and the probability density for $\mathbf{x}$ is $p(\mathbf{x}) = \frac{\mathcal{S}(\mathbf{x})}{\sum_{\mathbf{x}' \in \mathcal{X}} \mathcal{S}(\mathbf{x}')}$. They are members of the family of probabilistic circuits [Van den Broeck et al., 2019]. A special class, to be precise, that satisfies properties known as completeness and decomposability. Let $\mathsf{N}$ denote a node in SPN $\mathcal{S}$, then

$$\mathbf{sc}(\mathsf{N}) = \begin{cases} \{X\} & \text{if } \mathsf{N} \text{ is IV } (\lambda_{X=x}) \\ \bigcup_{\mathsf{C} \in \text{ch}(\mathsf{N})} \mathbf{sc}(\mathsf{C}) & \text{else} \end{cases} \tag{2}$$

is called the scope of $\mathsf{N}$ and

$$\forall \mathsf{S} \in \mathcal{S} : (\forall \mathsf{C}_1, \mathsf{C}_2 \in \text{ch}(\mathsf{S}) : \mathbf{sc}(\mathsf{C}_1) = \mathbf{sc}(\mathsf{C}_2)) \tag{3}$$

$$\forall \mathsf{P} \in \mathcal{S} : (\forall \mathsf{C}_1, \mathsf{C}_2 \in \text{ch}(\mathsf{S}) : \mathsf{C}_1 \neq \mathsf{C}_2 \implies \mathbf{sc}(\mathsf{C}_1) \cap \mathbf{sc}(\mathsf{C}_2) = \emptyset) \tag{4}$$

are the completeness and decomposability properties respectively. Since their introduction, SPNs have been heavily studied such as by [Trapp et al., 2019] that present a way to learn SPNs in a Bayesian realm whereas [Kalra et al., 2018] learn SPNs in an online setting. Several different types of SPNs have also been studied such as Random SPN [Peharz et al., 2020], Credal SPNs [Levray and Belle, 2020] and Sum-Product-Quotient Networks [Sharir and Shashua, 2018]) to name a few. For more details readers are referred to the survey of París, Sánchez-Cauce, and Díez [2020]. On another note, Gated or Conditional SPNs (CSPNs) are deep tractable models for estimating multivariate, conditional probability distributions $p(\mathbf{Y}|\mathbf{X})$ over mixed variables $\mathbf{Y}$ [Shao et al., 2019]. They introduce functional gate nodes $g_i(\mathbf{X})$ that act as a functional parameterization of the SPN's information flow and leaf distributions given the provided evidence $\mathbf{X}$.

**Causal Models.** A Structural Causal Model (SCM) as defined by Peters et al. [2017] is specified as $\mathfrak{C} := (\mathbf{S}, P_{\mathbf{N}})$ where $P_{\mathbf{N}}$ is a product distribution over noise variables and $\mathbf{S}$ is defined to be a set of $d$ structural equations

$$X_i := f_i(\text{pa}(X_i), N_i), \quad \text{where } i = 1, \dots, d \tag{5}$$

with $\text{pa}(X_i)$ representing the parents of $X_i$ in graph $G(\mathfrak{C})$. An intervention on a SCM $\mathfrak{C}$ as defined in (5) occurs when (multiple) structural equations are being replaced through new non-parametric functions $\hat{f}(\widehat{\text{pa}(X_i)}, \hat{N}_i)$ thus effectively creating an alternate SCM $\hat{\mathfrak{C}}$. Interventions are referred to as *imperfect* if $\widehat{\text{pa}(X_i)} = \text{pa}(X_i)$ and as *atomic* if $\hat{f} = a$ for $a \in \mathbb{R}$. An important property

of interventions often referred to as "modularity" or "autonomy"[1] states that interventions are fundamentally of local nature, formally

$$p^{\mathfrak{C}}(X_i \mid \mathrm{pa}(X_i)) = p^{\hat{\mathfrak{C}}}(X_i \mid \mathrm{pa}(X_i)) \,, \tag{6}$$

where the intervention of $\hat{\mathfrak{C}}$ occured on variable $X_k$ opposed to $X_i$. This suggests that mechanisms remain invariant to changes in other mechanisms which implies that only information about the effective changes induced by the intervention need to be compensated for. An important consequence of autonomy is the truncated factorization

$$p(V) = \prod_{i \notin S} p(X_i \mid \mathrm{pa}(X_i)) \tag{7}$$

derived by Pearl [2009], which suggests that an intervention $S$ introduces an independence of an intervened node $X_i$ to its causal parents. Another important assumption in causality is that causal mechanisms do not change through intervention suggesting a notion of invariance to the cause-effect relations of variables which further implies an invariance to the origin of the mechanism i.e., whether it occurs naturally or through means of intervention [Pearl et al., 2016].

A SCM $\mathfrak{C}$ is capable of emitting various mathematical objects such as graph structure, statistical and causal quantities placing it at the heart of causal inference, rendering it applicable to machine learning applications in marketing [Hair Jr and Sarstedt, 2021]), healthcare [Bica et al., 2020]) and education [Hoiles and Schaar, 2016]. A SCM induces a causal graph $G$, an observational/associational distribution $p^{\mathfrak{C}}$, can be intervened upon using the *do*-operator and thus generate interventional distributions $p^{\mathfrak{C};do(\dots)}$ and given some observations $\mathbf{v}$ can also be queried for interventions within a system with fixed noise terms amounting to counterfactual distributions $p^{\mathfrak{C}|\mathbf{V}=\mathbf{v};do(\dots)}$. To query for samples of a given SCM, the structural equations are being simulated sequentially following the underlying causal structure starting from independent, exogenous variables and then moving along the causal hierarchy of endogenous variables (i.e., following the causal descendants).

The work closest to our work is by Brouillard et al. [2020] although it solves the different problem of causal discovery. Causality for machine learning has recently gained a lot of traction [Schölkopf, 2019] with the study of both interventions [Shanmugam et al., 2015] and counterfactuals [Kusner et al., 2017] gaining speed. For more details the readers are referred to [Zhang et al., 2018].

## 3   Interventional SPNs

Now we are ready to develop interventional SPNs (iSPNs). To this end, we re-introduce the importance of adaptability of models to interventional queries and present a newly curated synthetic data set to both motivate and validate the formalism of iSPNs that, through over-parametric extension of SPNs, allows them to adhere to causal quantities.

### 3.1   Adaptation to Causal Change

Peters et al. [2017] motivated the necessity of causality via the adequate generalizability of predictive models. Specifically, consider a simple regression problem $f(a) = b$ with data vectors $\mathbf{a}, \mathbf{b} \in \mathbb{R}^k$ that are strongly positively correlated in some given region, e.g. $(1 < a < \infty, 1 < b < \infty)$. Now a query is posed outside the data support, e.g. $(0, f(0))$. As argued by Peters et al., the underlying data generating processes can be an ambiguous causal process i.e., the data at hand can be explained by two different causal structures being either $A \to B$ or a common confounder with $A \leftarrow C \to B$.

Assuming the wrong causal structure or ignoring it altogether could be fatal, therefore, any form of generalization out of data support requires assumptions to be made about the underlying causal structure. We adopt this point of view and further argue that *ignoring causal change(s) in a system, i.e., the change of structural equation(s) underlying the system, can lead to a significant performance decrease and safety hazards*[2]. Therefore, it is important to account for distributional changes present in the data due to experimental (thus interventional) settings.

---

[1]See Section 6.6 in [Peters et al., 2017].

[2]This extended notion of performance degeneration through ignorance to the underlying causality is prioritized in this paper.

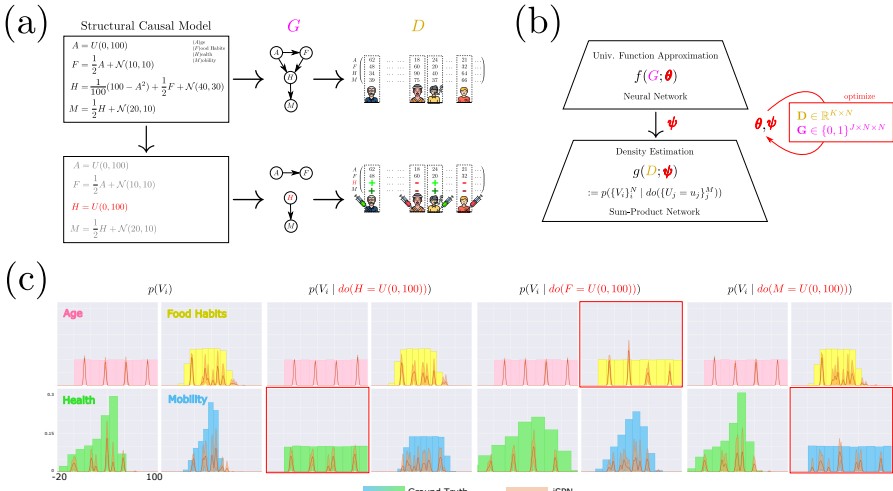

Figure 2: **An Overview of iSPN.** (a) The hidden process underlying the observable reality is modelled via a SCM that can be modified through interventions. The given SCM induces a causal graph and can generate data. An intervention can significantly alter the resulting data. (b) An over-parameterized density estimation framework using SPN is presented. The universal function approximator (here neural network) conditions on the mutilated causal graph and provides parameters to the SPN such that the given data's density can be modelled accordingly. The FA subsumes the $do$-operator. (c) Different causal queries are being presented. Furthermore, iSPN adapts to intervention-consequences.

Consequently, we consider the learning problem, where the given data samples have been generated by different interventions $do(\mathbf{U}_j = \mathbf{u}_j)$ in a common SCM $\mathfrak{C}$ while the induced mutilated causal graphs $G(\mathfrak{C}, do(\mathbf{U}_j = \mathbf{u}_j))$ are assumed to be known, such that the trained model is capable of at least inferring all involved causal distributions $p(V(\mathfrak{C}) \mid do(\mathbf{U}_j = \mathbf{u}_j))$ with $V$ being the variables.

### 3.2 Data Generating Process

To both validate and demonstrate the expressivity of interventional SPNs in modelling arbitrary interventional distributions, we curate a new causal data set based on the SCM $\mathfrak{C}$ presented in Fig. 2(a), which we subsequently refer to as *Causal Health* data set. The SCM encompasses four different structural equations of the form $V_i = f(\mathrm{pa}(V_i), N_i)$, where $\mathrm{pa}(V_i)$ are the parents of variable $V_i$ and $N_i$ are the respective noise terms that form a factor distribution $P^{N_{1:N}}$ i.e., the $N_i$ are jointly independent. Now, the SCM $\mathfrak{C}$ describes the causal relations of an individual's health and mobility attributes with respect to their age and nutrition.

Note that the Causal Health data set does not impose assumptions over the type of random variables or functional domains of the structural equations[3], which additionally constraints a learned model to adapt flexibly. While we generally do not restrict our method to any particular type of intervention, the following mainly considers perfect interventions as introduced in Sec. 2.

Perfect interventions fully remove the causal mechanism of the parents of a given node, which is consistent with the idea behind randomized controlled trials (RCTs) where the given intervention randomizes the given specific node, often referred to as gold standard in causality related literature. We consider the special case of uniform randomization, i.e., uniform across the domain of the given variable[4]. An intervention performed on one node immediately changes the population and, thus, has a major effect on the generating processes of subsequent causal mechanisms in the respective causal sequence of events.

To provide the reader with a concrete example of interventions within the causal health data set, consider the following: In concern of a virus infection the individuals of the Causal Health study should be vaccinated. The vaccine is expected to have side-effect(s), however, it has been poorly designed and has reached the population with the capability of completely changing the individual's

---

[3]Assumptions on the func. form of structural eq. are crucial for identification (Tab. 7.1 [Peters et al., 2017])
[4]Note that for binary variables this amounts to Bernoulli $B(\frac{1}{2})$.

health state. The observed changes do not show any form of pattern and are therefore assumed to be random. A young fit person could thus become sick, while an old person might feel better health wise. This sudden change will have an effect on the individual's mobility and also be independent of their age and nutrition. Such a scenario is mathematically being captured through $do(H = U(H))$ where $U(\cdot)$ is the uniform distribution over a given domain.

### 3.3 Introducing Interventional SPNs

After motivating both the importance and the occurrences of interventions within relevant systems, we now start introducing interventional SPNs (iSPNs).

**Definition of iSPN.** As motivated in Sec. 1, the usage of the compilation method from [Zhao et al., 2015] for causal inference within SPN is arguably of degenerate[5] nature given the properties of the compilation method [Papantonis and Belle, 2020]. While the results of Papantonis and Belle [2020] are arguably negative, there exists *yet no proof of non-existence of such a compilation method* and as the authors point out in their argument for future lines of research in this direction, a model class extension poses a viable candidate for overcoming the problems of using SPN for causal inference.

While agreeing on the latter aspect, we do not go the "compilation road" but extend the idea of conditional parameterization for SPN [Shao et al., 2019] by conditioning on a modified form of the *do*-operator introduced by Pearl [2009] while predicting the complete set of observed variables.

Mathematically, we estimate $p(V_i \mid do(\mathbf{U}_j = \mathbf{u}_j))$ by learning a non-parametric function approximator $f(\mathbf{G}; \boldsymbol{\theta})$ (e.g. neural network), which takes as input the (mutilated) causal graph $\mathbf{G} \in \{0, 1\}^{N \times N}$ encoded as an adjacency matrix, to predict the parameters $\boldsymbol{\psi}$ of a SPN $g(\mathbf{D}; \boldsymbol{\psi})$ that estimates the density of the given data matrix $\{\mathbf{V}_k\}_k^K = \mathbf{D} \in \mathbb{R}^{K \times N}$. With this, iSPNs are defined as follows:

**Definition 1** (Interventional Sum-Product Network). *An interventional sum-product network (iSPN) is the joint model $m(\mathbf{G}, \mathbf{D}) = g(\mathbf{D}; \boldsymbol{\psi} = f(\mathbf{G}; \boldsymbol{\theta}))$, where $g(\cdot)$ is a SPN, $f(\cdot)$ a non-parametric function approximator and $\boldsymbol{\psi} = f(\mathbf{G})$ are shared parameters.*

They are called interventional because we consider it to be a causal model given its capability of answering queries from the second level of the causal hierarchy [Pearl, 2019], namely, that of interventions[6]. The shared parameters $\boldsymbol{\psi}$ allow for information flow during learning between the conditions and the estimated densities. Setting the conditions such that they contain information about the interventions, in the form of the mutilated graphs $\mathbf{G}$, effectively renders $f$ to subsume a sort of *do*-calculus in the spirit of truncated factorization shown in Eq.(7) i.e., the gate model acts as an estimand selector. Generally, we note that our formulation allows for different function and density estimators $f, g$. We choose $f$ to be a neural network for two reasons (1) their empirically established capability to act as causal sub-modules (e.g. Ke et al. [2019] use a cohort of neural nets to mimic a set of structural equations, thus, a SCM) and (2) their model capacity being a universal function approximator, while we choose $g$ to be a SPN for its tractability properties for inference.

We have argued the importance of adaptability to interventional changes within the causal system and intend now to prove that iSPN are capable of approximating these different causal quantities.

**Proposition 1** (Expressivity). *Assuming autonomy and invariance, an iSPN $m(\mathbf{G}, \mathbf{D})$ is able to identify any interventional distribution $p_G(\mathbf{V}_i = \mathbf{v}_i \mid do(\mathbf{U}_j = \mathbf{u}_j))$, permitted by a SCM $\mathfrak{C}$ through interventions, with knowledge of the mutilated causal graph $\hat{G}$ and data $\mathbf{D}$ generated from the intervened SCMs by modelling the conditional distribution $p_{\hat{G}}(\mathbf{V}_i = \mathbf{v}_i \mid \mathbf{U}_j = \mathbf{u}_j)$.*

*Proof.* It follows directly from the definition of the *do*-calculus [Pearl, 2009] that $p_G(\mathbf{V}_i = \mathbf{v}_i \mid do(\mathbf{U}_j = \mathbf{u}_j)) = p_{\hat{G}}(\mathbf{V}_i = \mathbf{v}_i \mid \mathbf{U}_j = \mathbf{u}_j)$ where $\hat{G}$ is the mutilated causal graph according to the intervention $do(\mathbf{U}_j = \mathbf{u}_j)$ i.e., observations in the intervened system are akin to observations made when intervening on the system. Given the mutilated causal graph $\hat{G}$ (as adjacency matrix), the only remaining aspect to show is that the density estimating SPN can approximate a joint probability $p(\mathbf{X})$ using $\mathbf{D}$. This naturally follows from [Poon and Domingos, 2011]. $\square$

---

[5]A bipartite graph in which the actual variables of interest are not connected is called degenerate.
[6]The first level of the causal hierarchy —association— is considered to be purely statistical.

The expressivity of iSPN stems from both the capacities of gate function and the knowledge of intervention as well as availability of respective data. As an important remark, causal inference is often interested in estimating interventional distributions, i.e, causal quantities from purely observational models. Therefore, an alternative formulation to Prop. 1 would be to replace the knowledge of the intervened causal structure **G** with knowledge on a valid adjustment set. In the following, we only consider the direct setting where actual interventional data from the system is assumed to be captured[7] thereby freeing the investigation of iSPN from the independent research around hidden confounding.

**Universal Function Approximation (UFA).** The gating nodes of CSPNs extend SPNs in a way that allows them to also induce functions which are universal approximators. For instance, using threshold gates $x_i \leq c \in \mathbb{R}$, one can realize testing arithmetic circuits [Choi and Darwiche, 2018] which have been proven to be universal approximators - rendering iSPN to be UFA by construction.

**Learning of iSPN.** An interventional sum-product network is being learned using a set of mixed-distribution samples generated from simulating the Causal Health SCM for different interventions, where the observational case is considered to be equivalent to an intervention on the empty set. The parameters $\boldsymbol{\theta}, \boldsymbol{\psi}$ of the iSPN describe the weights of the gate nodes and the distributions at the leaf nodes. The full model $m$ is differentiable if the provided gate function $f$ and each of the leaf models of $g$ are differentiable. Therefore, to train an iSPN, as depicted in Fig. 2(b), it is sufficient to optimize the conditional log-likelihood end-to-end using gradient based optimization techniques. We assess the performance of our learned model through inspection of the adaptation of the model to the different interventions manifesting in the resulting marginals $p(V_i \mid do(U_j))$.

As can be observed in Fig. 2(c) or alternatively in Fig. 4 (top row), the learned iSPN successfully adapts to both the interventions as well as its consequences. Considering for instance the intervention $p(V_i \mid do(F = B(\frac{1}{2})))$ which removes the edge $A \rightarrow F$ and thus renders Age ($A$) and Food Habits ($F$) independent. Given the drastic population change in $F$ and the fact that the Health ($H$) of an individual is causally dependent on both $A$ and $F$ a significant change in $H$ is being expected. Indeed, both $H$ and Mobility ($M$), being a causal child of $H$, broaden distribution wise and also these subsequent changes are captured correctly.

**Causal Estimation.** The algebraic-graphical *do*-calculus [Pearl, 2009] is complete in that it can find estimands for any identifiable query in a finite amount of transformation. An SPN, and thereby also iSPN, is capable of providing estimates to said demands when given observational data ($\mathcal{L}_1$ on the PCH). Consider for instance the well-known non-Markovian "Napkin" graph $G = \{W \rightarrow Z \rightarrow X \rightarrow Y, W \overset{*}{\leftrightarrow} X, W \overset{*}{\leftrightarrow} Y\}$ where $\overset{*}{\leftrightarrow}$ denotes a confounded relation. The causal effect is identified as $p(y \mid do(x)) = (\sum_w p(x|w, z)p(w))^{-1}(\sum_w p(y, x|w, z)p(w))$ using the *do*-calculus where each of the r.h.s. components can be modelled by (i)SPN respectively. However, iSPN are also capable of directly expressing the causal quantity $p(y \mid do(x))$ as proven in Prop.1, similar to other neural-causal models like CausalGAN [Kocaoglu et al., 2019] or MLP-based NCM [Xia et al., 2021].

**Tractability.** Inference in BNs and Markov networks is at least an NP-problem and existing exact algorithms have worst-case exponential complexity [Cooper, 1990, Roth, 1996], while SPNs can do inference in time proportional to the number of links in the graph. I.e., SPNs in general are able to compute any marginalization and conditioning query in time linear of the model's representation size $r$, that is $O(r)$. We deploy simple neural network architectures for which the runtime for the forward-pass is that of matrix-multiplication i.e., the time complexity scales cubically in the size of the input $n$, that is $O(n^3)$. The gradient descent procedure that involves forward and backward passes, assuming $m$ gradient descent iterations, scales to $O(mn^3)$ which is the overall complexity we achieve during training phase. For SPN we deploy a random SPN structure which circumvents structure learning as such. Therefore, overall, training will generally be in the $O(mn^3r)$ regime while any causal query (that is, both $L_1$ observational and $L_2$ interventional in our case) will be answered within $O(n^3r)$. However, assuming that the weights of the random SPN were already initialized by the neural parameter-provider in a previous step, any causal query becomes answerable in $O(r)$. Since asking for inferences/queries $q$ within a single distribution ($q \rightarrow d \in L_i$) are more common than changes between distributions ($d, \hat{d} \subset L_i$), this linear complexity by our tractable model is being leveraged fully for performing causal inference.

**Discussion.** To reconsider and answer the general question of why the modelling of a conditional distribution via an over-parameterized architecture is a sensible idea consider the following. One can

---

[7]For details on the remarked alternative formulation consider the Supplement.

| Method \ Query | $V_1$ | $V_2$ | $V_3$ | $V_4$ |
|---|---|---|---|---|
| **iSPN** | $.001 \pm .00$ | $.007 \pm .01$ | $.003 \pm .00$ | $.013 \pm .01$ |
| **MADE** | $.588 \pm .59$ | $.108 \pm .16$ | $.015 \pm .02$ | $.105 \pm .12$ |
| **MDN** | $.178 \pm .14$ | $.263 \pm .14$ | $.184 \pm .12$ | $.079 \pm .01$ |

Table 1: **Jensen-Shannon-Divergence Evaluation of Estimated Interventional Distributions**. Numerical pendant to Fig.4, mean and standard deviation per $p(V_{j \setminus i} \mid do(V_i = U(V_i)))$ where $U$ is the uniform distribution across all data sets. Lower=better.

Figure 3: **Mean Running Times in sec. till convergence (Causal Health)** for 50 full passes. More data sets results in supplementary.

represent a conditional distribution $p(\mathbf{Y} \mid \mathbf{X})$ by applying Bayes Rule to a joint distribution density model (e.g. a regular SPN) $p(\mathbf{Y} \mid \mathbf{X}) = \frac{p(\mathbf{Y}, \mathbf{X})}{p(\mathbf{X})}$. However, this assumes non-empty support i.e., $p(\mathbf{X}) > 0$. Furthermore, the joint distribution $p(\mathbf{Y}, \mathbf{X})$ optimizes *all* possibly derivable distributions, diminishing single distr. expressivity. Therefore, our considered formulation of a gate model allows for effectively subsuming the $do$-operator i.e., the gate model orchestrates the $do$ queries such that the density estimator can easily switch between different interventional distributions. While not introducing specific limitations, general CSPN limitation regarding OOD generalization are inherited.

## 4    Experimental Results

The assumptions made in causality usually require control over the data generating process which is almost never readily available in the real world. This amounts to scarcity of the available public data sets and also their implications for transfers to the real world and even when available, they are usually artificially generated as some causal extension of a known, pre-existing data set (e.g. MorphoMNIST data set introduced by Castro et al. [2019]). While it is difficult to consider real-world-esque experimental settings for causal models, we do not restrict our investigations of iSPN to rather specific problem settings like certain noise or decision variable instances (which is common in causal inference). The evaluation is performed on data sets with varying number of variables and in both continuous and discrete domains. For the introduced causal health data set, we even consider arbitrary underlying noise distributions. We have made our code repository publicly available[8].

**Data Sets.** We evaluate iSPNs on four data sets. Three benchmarks: ASIA (A) [Lauritzen and Spiegelhalter, 1988] with 8 variables, Earthquake (E) with 5 variables and Cancer (C) [Korb and Nicholson, 2010] with 5 variables. One newly curated synthetic causal health (H) data set with 4 variables. More information about the data sets is presented in the Supplement.

**Baselines.** For *generative* capacities, we compare our method against Mixture Density Networks (MDN) [Bishop, 1994] and Masked Autoencoder for Density Estimation (MADE) [Germain et al., 2015]. Both methods are expressive, parametric neural network based approaches for density estimation. Generally, the causality for machine learning literature suggests a strong favor for neural based function approximators for modelling causal mechanisms [Ke et al., 2019]. For *causal* capacities, we compare our method against the renown causal baselines from CausalML [Chen et al., 2020] and DoWhy [Sharma and Kiciman, 2020] for the modelling of average treatment effects (ATE) [Pearl, 2009, Peters et al., 2017] considered to be a gold standard task within causal inference.

**Protocol and Parameters.** To account for reproducibility and stability of the presented results, we used learned models for five different random seeds per configuration. For means of visual clarity, the competing baselines MDN and MADE only present the best performing seed while iSPN is being presented with a mean plot and the standard deviation. The CBN, which performs exact inference according to the *do*-calculus (while best performing) has to be considered as gold standard and is therefore not part of the visualization. Furthermore, as it cannot compare in terms of feasibility. The considered interventions were of uniform nature. Each block of four columns represents a variable being randomized uniformly. We deployed a RAT-SPN [Peharz et al., 2020] selecting the leaf node distributions to be Gaussian distributions. For further experimental details consider the Supplement.

---

[8]https://github.com/zecevic-matej/iSPN

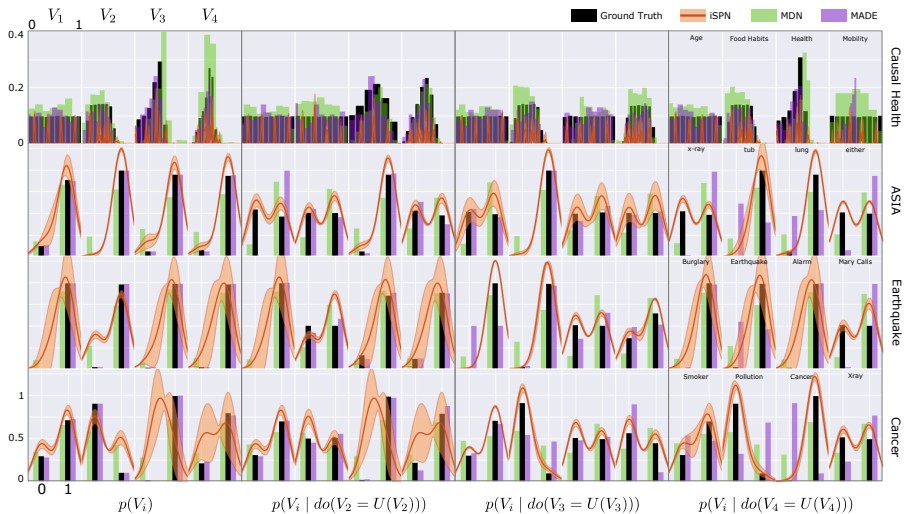

Figure 4: **Generative Baselines.** A comparison to the ground-truth (via underlying SCM) and competing estimated distributions. Each row represents a data set and each column represents a variable for a given causal query. ( Best in color.)

Our empirical analysis investigates iSPN for the following questions: **Q1**: How is the estimation quality for interv. distributions? **Q2**: How important is the model's capacity? **Q3**: How is the runtime performance? **Q4/5**: How is the performance relative to state-of-the-art generative and causal models? **Q6**: How does the model adapt to different types of interventions?

**(Q1. Precise Estimation of $do$-influenced variables)** The density functions learned by iSPN (see Fig. 4) fit with a high degree of precision as visualized by the difference in the peak of the modes of the learned distribution and that of the ground truth. While our visual analysis is arguably the superior method of evaluation as it conveys information about how the distributions of interest compare (which is feasible due to marginal inference and the locality of interventions within a SCM), we have also considered numerical evaluation criteria like the Jenson-Shannon-Divergence on which (as visually confirmed) iSPN outperforms the baselines (see Tab.1). For more detailed observation and interpretation consider the Supplement.

**(Q2. Capacity Ablation Study)** We test the robustness of iSPN as the size of the associated SPN $g(\mathbf{D}; \boldsymbol{\psi})$ is varied. We obtain 5 different iSPNs for each of the 4 data sets by using 5 different numbers of sum node weights, 600, 1200, 1800, 2400, 3200, effectively changing the capacity of the parameter-sharing neural network $f(\mathbf{G}; \boldsymbol{\theta})$. We observe iSPNs to be robust to varying hyper-parameters that control the size of the SPN $g(\cdot)$ and effectively the complexity of the associated function approximator $f(\cdot)$. More details and figures are in the Supplement.

**(Q3. Comparison of running times)** SPNs are tractable by design as long the networks size is polynomial in the input size. The superiority in running time also becomes apparent during training on the same (Causal Health) data set where we observe the mean run times over 50 passes of the whole data set to be significantly faster than competing methods (see Fig.3).

**(Q4. Comparison to Generative models)** We compare the performances in terms of precision of fit of the learned distributions as well as the flexibility of the models. iSPN outperforms the baselines across all 4 data sets both in precision of the fit of the learned density functions and flexibility of adaptation to the different interventions as seen in figure 4. In our experiments, MDN had the worst performance with estimated densities being consistently and significantly different from the ground truth, as the model settled for an average distribution across all interventions. MADE is able to estimate to high precisions but showed to be generally inconsistent across the experimental settings.

**(Q5. Comparison to Causal models)** We compare the numerical results as seen in figure 5 and observe that iSPN matches the performance of the causal baselines. The simple regressor employed by CausalML fails in the confounding case as it estimates wrongly the conditional, both DoWhy and iSPN can handle even the more difficult Simpson's paradox [Simpson, 1951] scenario. An analytical derivation for the ATE is exampled in the Supplement.

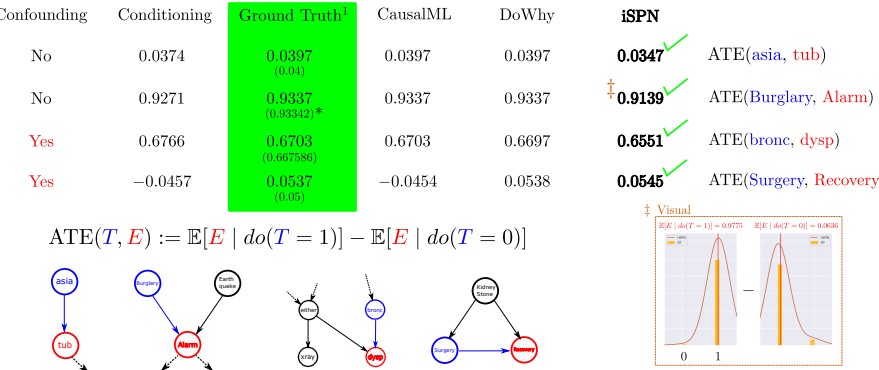

| Confounding | Conditioning | Ground Truth[1] | CausalML | DoWhy | iSPN | |
|---|---|---|---|---|---|---|
| No | 0.0374 | 0.0397 (0.04) | 0.0397 | 0.0397 | **0.0347** ✓ | ATE(asia, tub) |
| No | 0.9271 | 0.9337 (0.93342)* | 0.9337 | 0.9337 | ‡ **0.9139** ✓ | ATE(Burglary, Alarm) |
| Yes | 0.6766 | 0.6703 (0.667586) | 0.6703 | 0.6697 | **0.6551** ✓ | ATE(bronc, dysp) |
| Yes | −0.0457 | 0.0537 (0.05) | −0.0454 | 0.0538 | **0.0545** ✓ | ATE(Surgery, Recovery) |

$$\text{ATE}(T, E) := \mathbb{E}[E \mid do(T = 1)] - \mathbb{E}[E \mid do(T = 0)]$$

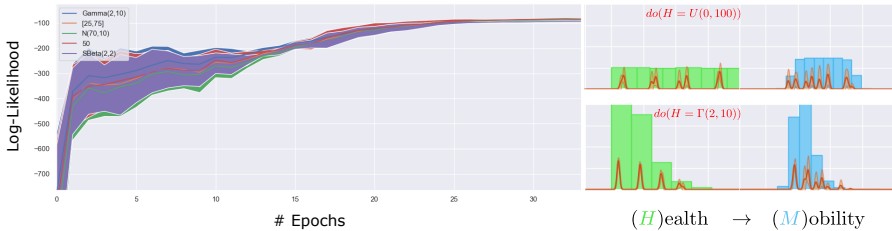

Figure 5: **Causal Baselines.** Different causal structures and corresponding causal effect estimation methods (CausalML, DoWhy) are being compared against iSPN. When confounding is present, then conditioning becomes different from intervening $p(Y \mid X) \neq p(Y \mid do(X))$ and iSPN correctly captures all evaluated cases. (* are analytical solutions, [1] differences of means for actual interventional distributions, Best viewed in color.)

Figure 6: **Adaptation to Different Interventions.** Training results for different kinds of interventions on the continuous CH data set. Left, the respective mean objective curves (log-likelihood), indicating consistent training and convergence for all three random seeds per configuration. Right, the (mean) density functions for two different interventions on $H$: Uniform $U(a, b)$ and Gamma $\Gamma(p, q)$ (other interventions shown in the supplementary). (Best viewed in color.)

**(Q6. Different Types of Intervention)** By construction, iSPNs are capable to handle arbitrary interventions and our empirical results corroborate this impression. Fig.6 shows the training of multiple models on different interventions alongside two example interventions and their appearance in the marginal distributions. Depending on the training setup, e.g. how many different interventional distributions need be learned, any single intervention might become more easily optimizable since the model can exploit similarities between distributions. For a more detailed elaboration and also visualizations of other interventional distributions (including atomic and non-continuous interventions), we direct the reader to our extensive supplementary material.

## 5 Conclusions

We presented a way to connect causality with tractable probabilistic models by using sum-product networks parameterized by universal function approximators in the form of neural networks. We show that our proposed method can adapt to the underlying causal changes in a given domain and generate near perfect interventional distributions irrespective of the data distribution and the intervention type thereby exhibiting flexibility. Our empirical evaluation shows that our method is able to precisely estimate the conditioned variables and outperform generative baselines.

Finding a different compilation method for SPNs such as by making use of tree CBNs is important for learning pure causal probabilistic models. Testing our method on larger real world causal data sets is an interesting direction. Finally, using rich expert domain knowledge in addition to observational data is essential for causality and extending our method to incorporate such knowledge is essential.

## Acknowledgments and Disclosure of Funding

The authors thank the anonymous reviewers for their valuable feedback. This work was supported by the ICT-48 Network of AI Research Excellence Center "TAILOR" (EU Horizon 2020, GA No 952215) and by the Federal Ministry of Education and Research (BMBF; project "PlexPlain", FKZ 01IS19081). It benefited from the Hessian research priority programme LOEWE within the project WhiteBox, the HMWK cluster project "The Third Wave of AI." and the Collaboration Lab "AI in Construction" (AICO) of the TU Darmstadt and HOCHTIEF. Furthermore, the authors gratefully acknowledge the support of 1R01HD101246 from NICHD and W911NF2010224 from US Army Research Office (ARO).

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
