# A    Appendix - Interventional Sum-Product Networks: Causal Inference with Tractable Probabilistic Models

We make further use of this supplementary section following the main paper to introduce some additional insights and results we deem important for the reader and for what has been examined in the main paper.

## A.1    Ablation Study on Arbitrary Intervention Realizations

In the main paper we have mostly considered perfect interventions i.e., interventions that render the intervened variables and its causal parents independent, and especially uniformly randomization as interventions which are consistent in their nature with the idea behind RCTs that are often argued to be the gold standard in causality. However, as already suggested, our model is not restricted to any specific intervention type or instantiation. Fig. 7 (a) illustrates the performance of iSPN on the Causal Health data set for different intervention types (perfect, atomic), noise terms (Gaussian, Gamma, Beta) and instantiations (Indicator Functions, Modifications). As can be observed, the model successfully manages to model most interventional distributions and consequences adequately. Furthermore, we observe that the training curves remained consistent among different intervention types further advocating the adaptability of the model to interventions of more arbitrary nature. Nonetheless, it can be observed that some interventions are being modelled more precisely than others, e.g. consider the relatively better performance of the model on the non-standard Beta intervention $do(H = 100B(2,2))$ that creates a wide and symmetric distribution opposed to the indicator intervention $do(H = 1_{[25,75]})$ that creates two heaps on 25 and 75. A possible explanation for this observation might lie in the fact that the model still learns other variants of distributions for a given variable, i.e. the different distributions the model adapts to on e.g. the Health $H$ variable are $p(H), p(H \mid do(F = f)), p(H \mid do(M = m))$ etc. and it can be argued that the non-standard Beta intervention is more consistent (that is, more similar) with these other marginal distributions of $H$ than it is with the unconventional distribution it learns for the indicator intervention. To summarize, the optimization problem becomes easier for the former.

## A.2    On the Visualization of Mean Densities

Another notable aspect for further informing the results from the main paper is to note that the visualization scheme, used in the main paper for assessing a given models performance, always considers learned models under multiple random seeds (i.e., their mean and standard deviation density functions) and never for a given single best seed. The natural motivation for the usage of multiple random seeds is to account for robustness and reproducibility within the results, however, a single best seed can significantly outperform the respective mean performance of a configuration, especially in regions of low data support which is not directly observable in the visualizations. Therefore, in Fig. 8 we show an example for the atomic intervention $do(H = 50)$. It can be observed that the single best seed fits the given ground truth perfectly without compromising for other learned distributions which does not become directly evident by observing the plotted mean performance. We argue that this is an important consideration to have in mind during visual inspection for adequate assessment of the observed results.

## A.3    Ablation Study on Function Approximator Capacity

As within the main paper, we perform an ablation study on the consequences of increasing or decreasing the function approximator (neural network) capacity for training any given configuration of iSPN. Fig. 11 visualizes the results for the remaining data sets not covered within the main paper i.e. Earthquake, Cancer and Causal Health. For Earthquake and Cancer data sets, we use 5 different number of sum node weights: 600, 1200, 1800, 2400 and 3200. For the synthetic causal health data set we use 300, 600, 1000, 1500, 2000. Each iSPN trained using these parameters is initialized over 5 random seeds. These values were chosen to test the performance of iSPN as they are increased/decreased compared to the optimal values of 2400 (600) for the public (synthetic) data sets. Interestingly, higher variances too, as seen in $p(V_3 \mid do(V_1 = U(V_1)))$, with $V_3$ being Xray and $V_1$ being Dyspnoea respectively, in the Cancer data set, are consistent across the 5 different iSPNs. The results are overall consistent with the presented i.e., the mean performance are consistent across the different neural network sizes while the variance can vary slightly on the presented data sets.

### A.4 Data sets for Generative and Causal Inference Tasks

Additional details on the four data sets considered for emprical evaluation of the generative and causal capabilities of iSPN in comparison to SotA methods:

| Data | # of variables | # of samples | # of edges |
|---|---|---|---|
| Causal Health | 4 | 100,000 | 4 |
| ASIA | 8 | 10,000 | 8 |
| Earthquake | 5 | 10,000 | 4 |
| Cancer | 5 | 10,000 | 4 |

Table 2: **Dimensions of the used data sets**. Edges refer to the edges in the causal graph associated with each data set. The benchmarks stem from: `https://www.bnlearn.com/bnrepository/discrete-small.html`

### A.5 Additional Experimental Details, Observation and Interpretation

Legend for Figure 4, data sets from top to bottom: H,A, E, C, variables from left to right: H: Age, Food Habits, Health, Mobility; A: Xray, Tub, Lung, Either; E: Burg., Earth., Alarm; C: Smoker, Poll., Cancer, Xray. We trained iSPNs on 10,000 samples for each of the 3 public data sets and on 100,000 samples for the synthetic Causal Health data set. We chose the non-parametric function approximators $f(\cdot)$ for each iSPN to be a multi-layer perceptron (MLP). The inputs to the MLPs were mutilated causal graphs $\hat{G}$. The MLPs had 2 hidden layers consisting of 10 units each and used ReLU as their activation functions. The outputs were 2400 (600) weights corresponding to the sum nodes and 96 (12) weights corresponding to the leaf nodes of the SPN $g(\mathbf{D}; \boldsymbol{\psi})$ for ASIA, Cancer and Earthquake (Causal Health) data sets. The MLPs were trained for 20 (130) epochs with a batch size of 100 (1000) and 5 different seeds.

Also, the learned distributions are similar across different seeds, with exception of some distributions, such as the marginal $p(V_4 | do(V_2 = U(V_2)))$ in the Cancer data set. A possible explanation for the observed higher variance is that the optimization trajectories during training for the different random seeds deviate with similar variance, stemming from the fact that different seeds select different initializations of the neural network parameters $\boldsymbol{\theta}$ leading to different optimization steps and possibly local optima. A hint to this deviation of optimization trajectories might also be the observed discrepancy between the single best seed of a given model configuration and its mean performance across multiple random seeds as is being pointed out to the reader in the Appendix A.2.

All of the experiments have been conducted on a MacBook Pro (13-inch, 2020, Four Thunderbolt 3 ports) laptop running a 2,3 GHz Quad-Core Intel Core i7 CPU with a 16 GB 3733 MHz LPDDR4X RAM on time scales ranging from seconds to (a few) hours with increasing size of the experiments.

### A.6 Alternative Formulation with an Adjustment Set

The following is an alternative to Prop.1 such that one exchanges the necessity of intervention with the knowledge on confounders if available.

**Proposition 2** (Adjustment for Observational Models). *Assuming autonomy and invariance, any interventional distribution $p_G(\mathbf{V}_i = \mathbf{v}_i \mid do(\mathbf{U}_j = \mathbf{u}_j))$ permitted by a SCM $\mathfrak{C}$ that induces a causal graph $G$ can be identified by adjusting for the confounders $\mathbf{w}$ within the observational model $\sum_{\mathbf{w}} p(\mathbf{V}_i = \mathbf{v}_i \mid \mathbf{U}_j = \mathbf{u}_j, \mathbf{W} = \mathbf{w}) p(\mathbf{W} = \mathbf{w}).$*

*Proof.* Assume $\mathbf{W} \cap \{\mathbf{V}, \mathbf{U}\} = \emptyset$. Furthermore, $G, \hat{G}$ are again the original and intervened causal graph and $p = p_G$. Now, $p(\mathbf{V}_i = \mathbf{v}_i \mid do(\mathbf{U}_j = \mathbf{u}_j)) = p_{\hat{G}}(\mathbf{V}_i = \mathbf{v}_i \mid \mathbf{U}_j = \mathbf{u}_j)$

$$= \sum_{\mathbf{w}} p_{\hat{G}}(\mathbf{V}_i = \mathbf{v}_i, \mathbf{W} = \mathbf{w} \mid \mathbf{U}_j = \mathbf{u}_j)$$

$$= \sum_{\mathbf{w}} p_{\hat{G}}(\mathbf{V}_i = \mathbf{v}_i \mid \mathbf{U}_j = \mathbf{u}_j, \mathbf{W} = \mathbf{w}) p_{\hat{G}}(\mathbf{W} = \mathbf{w})$$

$$= \sum_{\mathbf{w}} p(\mathbf{V}_i = \mathbf{v}_i \mid \mathbf{U}_j = \mathbf{u}_j, \mathbf{W} = \mathbf{w}) p(\mathbf{W} = \mathbf{w}) \, .$$

The first equality follows, by definition, from *do*-calculus as argued in Prop. 1, i.e., the intervention amounts to the observation in the intervened system. The second and third equality are transformations

according to the rules of probability theory: the second step follows the sum rule and the third step follows the chain rule. The last line follows from the assumptions of autonomy and invariance i.e., that interventions are local and invariant to whether they occur naturally or artificially. $\qquad\square$

The set of variables $\mathbf{W}$ is called *(valid) adjustment set* (see Def. 6.38 [Peters et al., 2017]) if it adjusts the observational setting such that the causal effect captured by the intervention becomes *unconfounded*[9]. For the Causal Health SCM $\mathfrak{C}$ which induces the simple causal graph $A \to F, \{A, F\} \to H, H \to M$, the equivalence of Props. 1 and 2 depends on the inference query. While an intervention on $A$ is trivially unconfounded $p(H = h \mid do(A = a)) = p(H = h \mid A = a)$, an intervention on $F$ would require adjustment via e.g. $\{A\}$, that is, $p(H = h \mid do(F = f)) = \sum_a p(H = h \mid F = f, A = a)p(A = a)$.

### A.7 Example of an Analytical Derivation for ATE

Consider the non-confounding case of the ATE or causal effect of a Burglary $(B)$ on an Alarm $(A)$ (where $Q$ is Earthquake) from the Earthquake data set. From the law of iterated expectations, it follows:

$$
\begin{aligned}
&ATE(\text{Treatment=}B, \text{Effect=}A) \\
&= \mathbb{E}[A \mid do(B = 1)] - \mathbb{E}[A \mid do(B = 0)] \\
&= \mathbb{E}_Q[\mathbb{E}[A \mid do(B = 1), Q] - \mathbb{E}[A \mid do(B = 0), Q]] \\
&= 0.99322 - 0.0598 = \mathbf{0.93342}
\end{aligned}
$$

with

$$
\begin{aligned}
&\mathbb{E}_Q[\mathbb{E}[A \mid do(B = 1), Q]] \\
&= \sum_e \mathbb{E}[A \mid do(B = 1), Q = e]p(Q = e) \\
&= \sum_e p(A = 1 \mid do(B = 1), Q = e)p(Q = e) \\
&= \sum_e p(A = 1 \mid B = 1, Q = e)p(Q = e) \\
&= 0.71 * 0.02 + 0.999 * 0.98 = 0.99322
\end{aligned}
$$

and analogously for $\mathbb{E}_Q[\mathbb{E}[A \mid do(B = 0), Q]]$.

The ATE tells us that a burglary has a strong effect on the alarm to be triggered which is consistent with our human intuition as the alarm is specifically designed to trigger in an event of theft.

---

[9]Mathematically, the causal effect from $X$ to $Y$ is confounded if $p(Y \mid do(X)) \neq p(Y \mid X)$

## A.8 Further Numerical Evaluation

More numerical evaluation, using Jensen-Shannon-Divergence as the discrepancy measure for the simulated ground truth distribution against the estimates of iSPN and the generative competition. A lower score suggests a better approximation. We consider a cycle permuted experimental setup in which each experiment considers a different uniform intervention of the currently selected variable of choice. Generally, iSPN outperforms the baselines in its estimate precision under fair (i.e., similar neural capacities, same amount of data passes, same amount of random seeds under consideration):

| Method / Query | iSPN | MADE | MDN | Method / Query | iSPN | MADE | MDN |
|---|---|---|---|---|---|---|---|
| x-ray | 0.002 | 1.262 | 0.398 | Burglary | 0.000 | 1.083 | 0.182 |
| tub | 0.009 | 0.004 | 0.176 | Earthquake | 0.000 | 0.005 | 0.465 |
| lung | 0.002 | 0.003 | 0.285 | Alarm | 0.001 | 0.048 | 0.312 |
| either | 0.009 | 0.022 | 0.089 | MaryCalls | 0.005 | 0.033 | 0.084 |

| Method / Query | iSPN | MADE | MDN | Method / Query | iSPN | MADE | MDN |
|---|---|---|---|---|---|---|---|
| Smoker | 0.003 | 0.002 | 0.064 | Age | 0.000 | 0.003 | 0.069 |
| Pollution | 0.005 | 0.385 | 0.305 | Food Habits | 0.013 | 0.037 | 0.108 |
| Cancer | 0.003 | 0.002 | 0.064 | Health | 0.006 | 0.007 | 0.073 |
| Xray | 0.001 | 0.305 | 0.066 | Mobility | 0.037 | 0.060 | 0.079 |

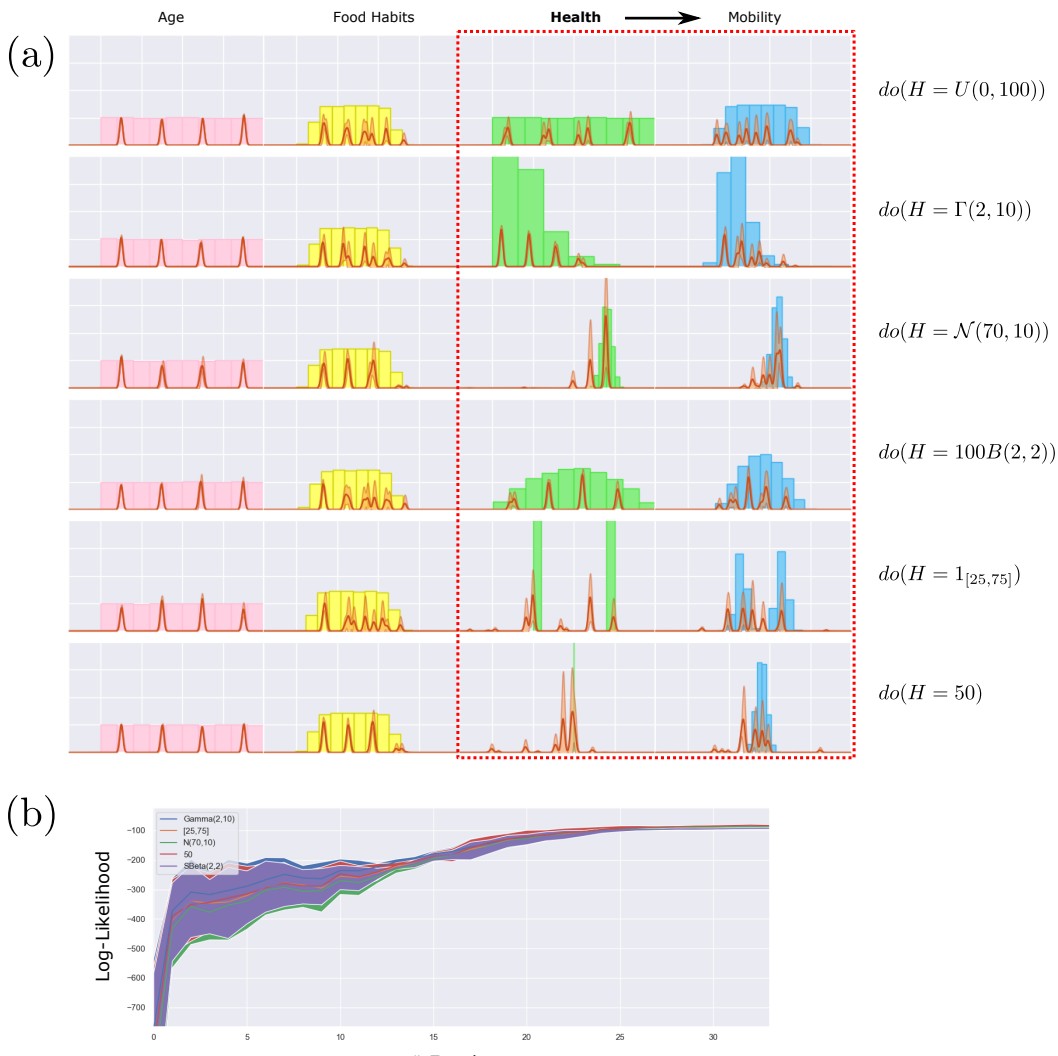

Figure 7: **Ablation Study on Different Intervention Types, Noise Terms and Instantiations.** Training results for different kinds of interventions on the continuous Causal Health data set are being presented, with (a) presenting the learned (mean) density functions for a given intervention on $H$, where we consider different noise distributions normal $\mathcal{N}(\mu, \sigma^2)$, Gamma $\Gamma(p, q)$, and Beta $B(a, b)$ but also different modifications, e.g. the non-standard Beta distribution $(k-l)B(a, b)+l$ and intervention types, e.g. perfect interventions $do(H = a), a \in \mathbb{R}$. Below (b) shows the respective mean objective curves (log-likelihood), indicating consistent training and convergence on the continuous data set for all 3 seeds per configuration.

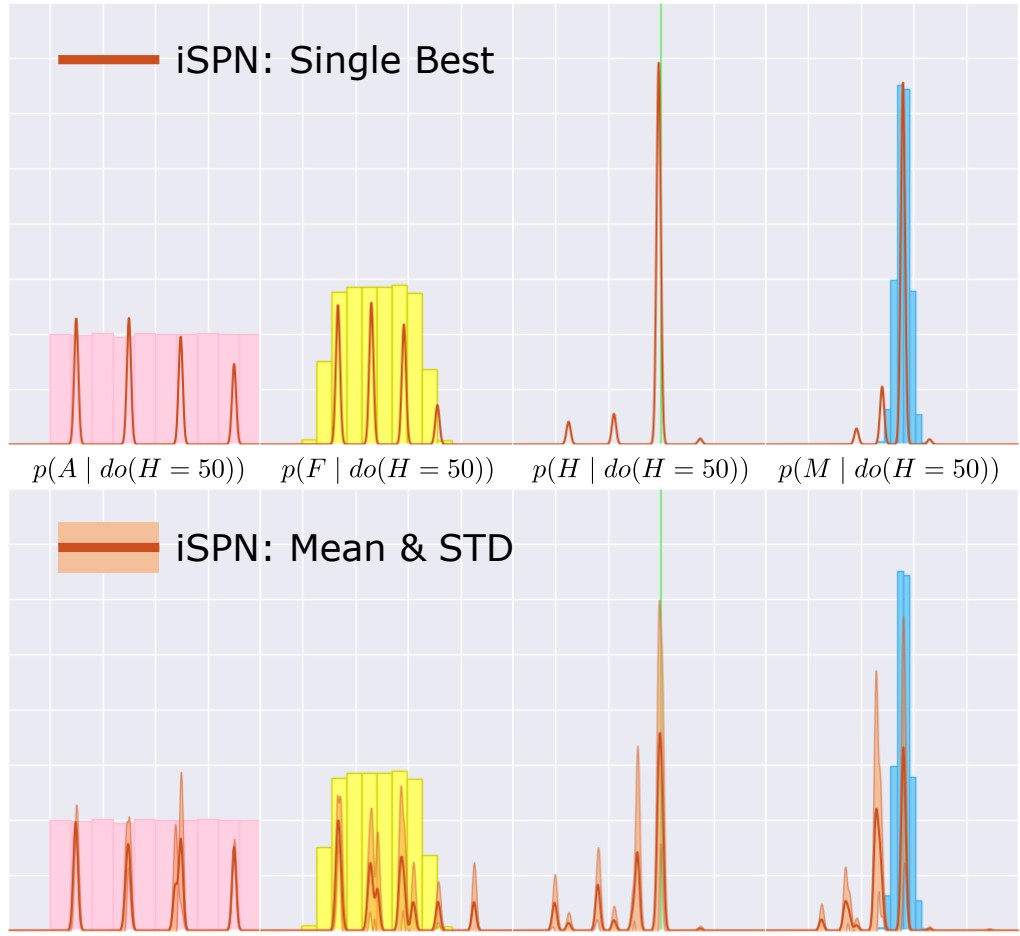

Figure 8: **Single Best Density vs Mean Density.** An example of the visual discrepancy between the performance of the single best seed and the mean performance across multiple seeds. As can be observed in the top row, the single best seed fits all the different marginal distributions accurately, while the mean performance of the given training setup and model architecture (presented in the bottom row) shows deviation especially on regions of low or none support i.e., consider $p(H \neq 50 \mid do(H = 50)) > 0$ that have more emphasized peaks although ideally none (or insignificant ones, considerably noise, as for the single best seed) should be observed.

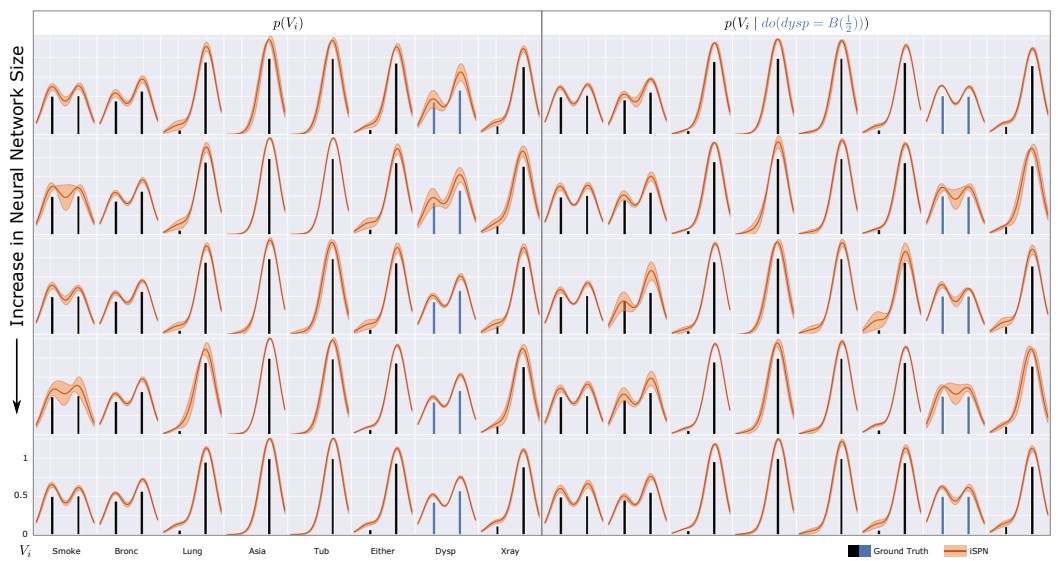

Figure 9: **Effect of size of neural network.** We test robustness of iSPN by training with neural networks of 5 different parameter sizes on the ASIA data set. Each row represents neural network size, the first 8 columns represent the observational distributions of 8 variables and the next 8 columns represent the learned marginal distributions upon intervention with $do(\cdot)$.

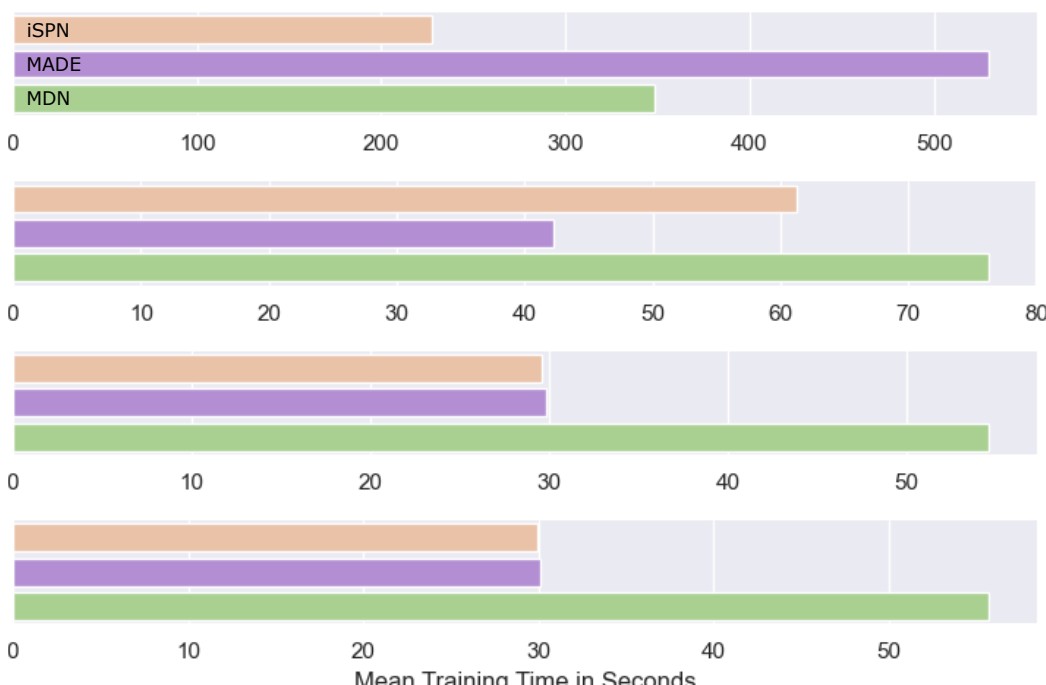

Figure 10: **Runtimes over different data sets.** From top to bottom, as in Fig.4: Causal Health (H), ASIA (A), Earthquake (E), Cancer (C). On the binary data sets (A, E, C), MADE due to its single-model form is able to match iSPN and slightly improve upon iSPN, however, for the hybrid-model real-world-esque (H) the speed advantage of the over-parameterized models and especially iSPN becomes apparent. Important: note the scales on H, iSPN significantly outperforms the other models.

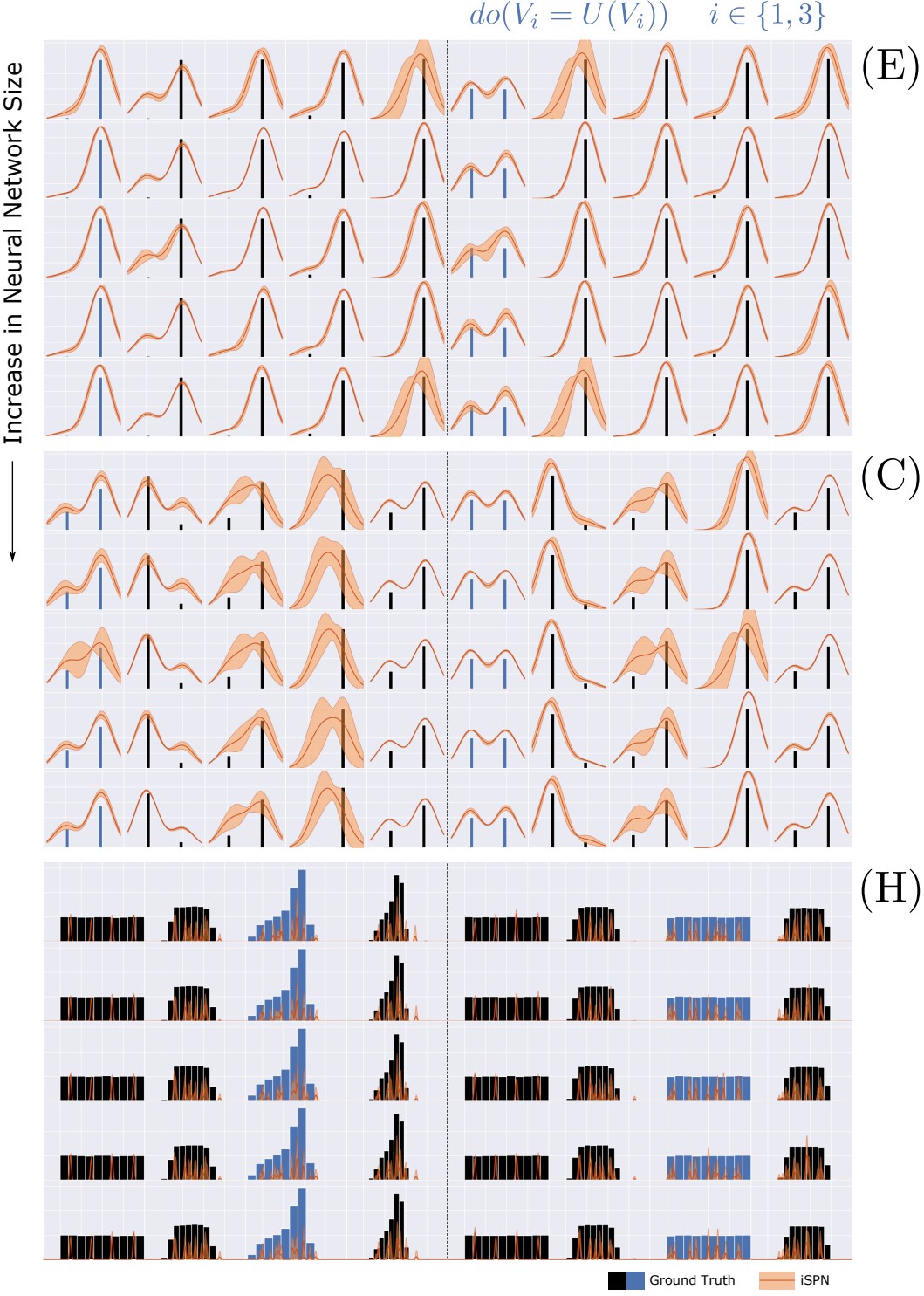

Figure 11: **Ablation Study on Influence of Neural Network Size.** A cohort of trained model configurations that only differ in the size of their respective function approximation modules (neural network) and their mean performances across the observational and one interventional setting (uniform intervention being Bernoulli $B(\frac{1}{2})$ for discrete and Uniform $U(V_i)$ for continuous variables) are being presented for the remaining data sets: Earthquake (E), Cancer (C), Causal Health (H). As can be observed, the mean performance stays mainly consistent across the different function approximation capacities while the variance can differ slightly.