# OpenReview forum: "Interventional Sum-Product Networks: Causal Inference with Tractable Probabilistic Models"
_NeurIPS.cc/2021/Conference — NeurIPS 2021 Poster_

### Official Review · Reviewer_mLY8 · 2021-07-12

**Rating:** 7
**Confidence:** 3

**Summary:**

This paper discusses a variant of sum-product networks (SPNs) called "interventional sum-product networks" which is a variant of SPNs that take as input a causal graph and use this to model interventional distributions. The claim is that this is the first paper that applies tractable probabilistic models to causality. Theoretical aspects are discussed and the method is evaluated and compared to existing methods.


**Limitations And Societal Impact:**

The societal impact is not relevant for this paper. There is a brief discussion on limitations, in context of future work. This seems appropriate.

**Main Review:**

The clarity of the paper could be improved. The key definition in the paper is the definition of an interventional SPN (Definition 1), which is defined very abstractly, i.e., the concepts SPN and gated SPN are not discussed in detail. It is also not clear to me why f(G) are shared parameters (shared between what?). There is also quite some discussion on SCMs, but the paper never really makes clear what kind of SCMs the method supports. For example, the paper does not say whether the SCM should be acyclic (and whether cyclic SCMs were evaluated).  Besides that, the paper is well-structured and quite easy to read. Improving the clarity slightly should be little work.

As far as I know, this is a novel approach of combining causal knowledge as part of a "deep" representation that can be learned from data, bridging a gap between two areas that usually do not have a lot of overlap. While in this paper the authors chose SPNs, but it is not hard to see that similar ideas could be applied in other areas. As a result, I think the paper could have significant impact, and could open new avenues for research.

Technically, the paper is straightforward and seems correct. The experiments are limited, but seem appropriate for this paper: I think it convincingly shows that iSPNs are capable for learning generative models and provides good causal effect estimations. In my opinion, this is sufficiently convincing.

**Time Spent Reviewing:**

2

---

> ### Author Response · Authors · 2021-08-10
> **Author's response**
>
> Thank you for your very positive review, $\color{purple}{\text{reviewer mLY8}}$!
>
> * > $\color{purple}R:$ "I think the paper could have significant impact, and could open new avenues for research"
>
>   We feel delighted by this assessment which we hope does hold for the future and we also strongly agree with the "new avenues" for research aspect, and in fact have in the meantime collected some preliminary results regarding this.
>
> * > $\color{purple}R:$ "the paper is well-structured and quite easy to read. Improving the clarity slightly should be little work"
>
>   A good structure and accessibility have been some of our key goals when writing this paper. Regarding clarity, thank you for your suggestion if accepted we will cover the points to as discussed in the two bullet points following below which reference it in the context of two of your key questions.
>
> * > $\color{purple}R:$ "not clear to me why f(G) are shared parameters (shared between what?)"
>
>   We have observed this to be a key question of the review. We actually refer to a different meaning of the word sharing and that we can make explicit in the following way: In the context of Deep Leaning, parameter sharing might usually refer to a set of layers in a model which will all use the same parameters. But in our case we just wanted to express the connectivity between models, but the parameters are not being used by the NN they are just being 'shared' with the SPN (see p.6,l.193f.). This is also what we hoped to actually make clear with our overview in Fig.2b.
>
> * > $\color{purple}R:$ "clear what kind of SCMs the method supports. For example, the paper does not say whether the SCM should be acyclic (and whether cyclic SCMs were evaluated)."
>
>   Another key concern was regarding the nature of SCMs. We support the general definition set in the book from [Peters et al. 2017 (see Definition 6.2)](https://library.oapen.org/bitstream/handle/20.500.12657/26040/11283.pdf?sequence=1&isAllowed=y), which actually assumes an acyclic structure (to quote the exact reference: "*The graph* $G$ of an SCM is obtained by creating one vertex for each $X_j$ and drawing directed edges from each parent in $PA_j$ *to $X_j$, that is, from each variable $X_k$* occurring on the right-hand side of equation (6.1) to $X_j$ *(see Figure 6.1). *We henceforth assume this graph to be acyclic*.*"). Therefore, no, we have not tested our method on such systems. But surely this is interesting! Because we don't make any assumptions on acyclicity in *our model*, so potentially there is something worthwhile to be gained here!
>
> * > $\color{purple}R:$ "the concepts SPN and gated SPN are not discussed in detail."
>
>   We acknowledge the feeling that it might seem short but we disagree in that we think that SPN and Gated SPNs are being adequately discussed i.e., as much as needed for the scope of the paper while respecting outer restrictions. We will consider adding an extended more detailed treatise to the appendix.
>
> * > $\color{purple}R:$ "the experiments are limited but seem appropriate for this paper"
>
>   Your assessment completely correctly points out the difficulty (and our feelings as researchers) of nowadays research around joining causality with modern machine learning when it comes to empirics - to which we surely hope to contribute to with our work.

---

### Official Review · Reviewer_aP4W · 2021-07-13

**Rating:** 3
**Confidence:** 4

**Summary:**

This submission proposes interventional sum-product networks, a tractable method for estimating causal effects from interventional data. The authors prove that iSPNs are universal function approximators. They show that iSPNs produce accurate estimates of interventional queries from interventional data on collection of synthetic data generating processes.

**Ethical Concerns:**

None.

**Limitations And Societal Impact:**

The authors discuss some limitations (in the form of future work) in the conclusions section.

**Main Review:**

While the authors begin to address an important general topic of tractable probabilistic causal inference, the current submission requires substantial revision before it is ready for publication. Specifically, it is unclear how the specific task the authors have focussed on is significant, the clarity of the writing could be improved, and the submission appears to be missing a technical description of the key methodology.

Significance of the Task:
As the authors note in the submission that "observations in the intervened system are akin to observations made when intervening on the system". In other words, inferring the effect of an intervention using data sampled from the same intervention reduces to standard probabilistic inference. In that vein, it appears that iSPNs are just SPNs using interventional data, and do not leverage causal assumptions or interventional semantics to relate observational data (or data from a different interventional setting) to inform estimates of interventional queries.

It is also not clear from the submission how the tractability of SPNs contribute to answering interventional queries from a learned model. As the submission only considers causal graphs without latent confounders, probabilistic inference in the (fully parameterized) intervened model reduces to forward sampling, which is already tractable for directed graphical models.

Clarity:
Given that the paper is assuming a known graph structure, it is unclear what relevance Section 3.1 has to the central claims and contributions. I suggest removing the discussion about how multiple causal structures induce the same observational distribution. In addition, the author appears to be conflating causal inference outside of the data support (i.e. violating positivity assumptions) with structural ambiguity.

My understanding of SPNs is that they result in tractable inference because they place restrictions on the space of joint densities that can be represented. However, this seems to contradict the submission's claims that iSPNs are universal function approximators. How do the authors reconcile this apparent contradiction?

Missing Presentation:
Section 3.3 describes what iSPNs are not, but does not thoroughly describe what iSPNS are. For example, how is g(*) constructed? This section could be substantially improved by including an algorithm block for each of the following; (i) how g(*) is constructed, (ii) how neural network components relate to underlying graph structure, and (iii) how iSPNs perform inference at query time. I understand that some of these will be very simple, but being explicit will help readers understand what is novel about the contribution, and what is a simple reduction of existing techniques.

Minor suggestions:

The figures could be substantially improved by clearly labeling and captioning subfigures. Additionally, some of the figure axis labels, etc. are too small to be legible.

Given that U is often used to denote latent confounders or exogenous noise in SCMs, I recommend choosing a different notation than do(U=u) throughout.


**Time Spent Reviewing:**

3

---

> ### Author Response · Authors · 2021-08-10
> **Author's response**
>
> We appreciate that you took your time reviewing our paper, $\color{purple}{\text{reviewer aP4W}}$!
>
> * > $\color{purple}R:$ "how the tractability of SPNs contribute to answering interventional queries from a learned model"
>
>   This is an excellent question! We deem this one more difficult and an advanced question which naturally follows from the work presented in our paper i.e., how causality and tractibility of probabilistic circuits (like SPNs) can be integrated more tightly within their core mathematical principles. Our thoughts are that the process of information/computation flow might be more deeply related to the concept of intervention.
>
> * > $\color{purple}R:$ However, this seems to contradict the submission's claims that iSPNs are universal function approximators. How do the authors reconcile this apparent contradiction?
>
>   This seemed to be a major concern and we will gladly point to where we handle this in the paper because this is an important point we in fact do cover adequately. We dedicate a whole paragraph (p.7,l.221-224) to the aspect of universal function approximation (UFA). In fact, gated or conditional SPNs are UFAs. Intuitively, one would assume because they can use neural networks and because neural nets are UFA, the gated/conditional SPNs become as well. However, Choi and Darwiche (2018) proved that already arithmetic circuits (ACs) are sufficient (neural nets are not needed!) for having a gated/conditional SPN to be UFA. Now, since iSPN are constructed on the base of gated/conditional SPN they are also UFA.
>
> * > $\color{purple}R:$ assuming a known graph structure, it is unclear what relevance Section 3.1 has to the central claims and contributions. I suggest removing the discussion about how multiple causal structures induce the same observational distribution.
>
>   We acknowledge the importance of pushing only relevant sections for the central claims of our contribution. However, we have concluded that subsection 3.1 is very important to the central claims of our paper as it sets the stage for the formulation of iSPN later in this section. Understanding the importance and key concepts behind interventions is crucial and not easily understood by everyone within the community, especially not by people who have yet not come in touch with causality.
>
> * > $\color{purple}R:$ In that vein, it appears that iSPNs are just SPNs using interventional data, and do not leverage causal assumptions or interventional semantics to relate observational data (or data from a different interventional setting) to inform estimates of interventional queries.
>
>   We disagree with the reviewer but this seems to be a very important concern and we would like to address this thoroughly. In fact, we think that it is possible that you have missed two key aspects of work that help in providing a satisfactory answer. The aspect regarding "are just SPNs" is more than what it seems as it is a very big open research question on which [Papantonis and Belle (2020)](https://arxiv.org/abs/2001.10905) seem to have a big opinion on. They generally conclude with a negative result (i.e., no causality for SPNs basically) but are not able to prove it (please see in our paper p.6,l.175-180). This was actually one of our key motivations for the paper, as we saw a chance in improving this by making use of deep leaning! To be more precise (and this is now the second key point to which our work contributes to), to make this meta-operator $do(\cdot)$ introduced by Judea Pearl a differentiable function like a neural net (for reference to this in our paper, please consider p.6,l.181-183).
>
> * > $\color{purple}R:$ by including an algorithm block for each of the following; (i) how g() is constructed, (ii) how neural network components relate to underlying graph structure, and (iii) how iSPNs perform inference at query time.
>
>   We certainly agree with you that being explicit and clear is important but that's exactly why we also have to disagree in that we don't think algorithm blocks would do justice for aspect (i). We think our choices are adequate and glady point to those in the paper, so for (i) please see Fig.2b in the main text, in the supplement p.2,l.58-64, and please also consider [our code that we made sure is reproducible over new setups](https://anonymous.4open.science/r/8a12a810-3343-4950-a5de-bd0721f4e914/README.md). Regarding aspect (ii) we have not yet investigated this surely interesting direction but we cover something similar in that we look at the relation of the general size of the neural networks to the subsequent model (see Supplement p.2, l.38-50). Regarding aspect (iii) there is no difference to standard procedures for gated/conditional SPNs but as you pointed out, especially to readers unfimiliar with this, it might be of great benefit, thereby we will consider this for the final version.
>
> **Our comments on the assessment of $\color{purple}{\text{reviewer aP4W}}$**:
>
> * We appreciate the time the reviewer took for us and evidently the value it generated when one looks at our comments to your review from above's bullet points. However, we have one issue with the review. The review did *not make explicit any positive points or strengths of our paper.*

---

### Official Review · Reviewer_3eLf · 2021-07-16

**Rating:** 5
**Confidence:** 4

**Summary:**

This paper proposes a modification of sum product networks termed iSPN to estimate interventional distributions from data. The proposed model performs a manipulation to the structure of the SPN in order to provide estimates from the mutilated distribution. The authors motivate the use of this model in terms of computational complexity and the ability to represent estimates of arbitrary causal quantities on the graph. Empirical results show the proposed model having impressive running time and comparable error results to modern software packages for causal effect estimation.

**Limitations And Societal Impact:**

It would be very helpful if the authors more clearly define the assumptions, limitations, and expected failure points of the proposed method.

**Main Review:**

I am of two minds on this paper. On one hand, I think there is real novelty in a black box representation such as what the authors are presenting. The proposed framework is very general and could likely be of real utility for developing a wide range of downstream applications. On the other hand this paper is overly vague in many places making it difficult to propose acceptance. The authors make use of "universal approximators" but don't provide any theory which would define the criteria for consistent and unbiased estimation from the proposed model. This is crucial in causal inference applications, which are essentially unsupervised models. It's also not clear to me what the comparative advantage is of this model over existing estimation procedures which come with _very_ strong theoretical guarantees and robust empirical performance. The authors point to computational efficiency, however computational efficiency is rarely the bottleneck for causal estimation tasks (whereas statistical efficiency often is). It is also unclear to me how hyperparameter selection is to be carried out. Finally, the authors don't make plain what assumptions are being required of the data in order to make valid inferences.

**Time Spent Reviewing:**

3

---

> ### Author Response · Authors · 2021-08-10
> **Author's response**
>
> Thank you for your sincere review, $\color{purple}{\text{reviewer 3eLf}}$, we see that you like our paper and that you are yet still of "two minds on this paper" to quote your words, therefore, we hope to clarify!
>
> * > $\color{purple}R:$ vague in many places making it difficult to propose acceptance.
>
>   We realized that while making your review you got the impression of vagueness at places within the paper which basically seems to be your only issue with our work. Gladly you covered these "vague" spots within your review such that we could identify them! Therefore, in the following two bullet points we cover the two main clusters of your questions thoroughly.
>
> * > $\color{purple}R:$ "universal approximators" but don't provide any theory
>
>   We dedicate a whole paragraph to this topic (please see p.7,l.221-224) and to sum it up concisely: the trick for iSPN lies in realizing that by construction they inherit their UFA properties from gated or conditional SPNs. Basically, adding such a discussion would be of no value other than repitition for our paper, however, if you are more interested, then we are gladly pointing you to the two of the most important works regarding this: [Shao et al. 2020](https://ml-research.github.io/papers/shao2020infspek_argML.pdf), [Darwiche et al. 2018](https://arxiv.org/abs/1812.08957).
>
> * > $\color{purple}R:$ It's also not clear to me what the comparative advantage is of this model over existing estimation procedures which come with very strong theoretical guarantees and robust empirical performance. The authors point to computational efficiency, however computational efficiency is rarely the bottleneck for causal estimation tasks (whereas statistical efficiency often is)
>
>   This questions of yours is regarding both theoretical guarantees of our method and scaled causal inference. For the first, the situation is in fact very similar to the bullet point above for which both references still hold but also consider the theoretical results within [this overview by Paris et al. 2020](https://arxiv.org/abs/2004.01167) and [Section 2 within Peharz et al. 2020](https://arxiv.org/pdf/2004.06231.pdf) additonally. To sum up, our underlying components come with theoretical guarantees and thus the whole system is theoretically sound. For the second part, we have to respectfully disagree in the view that computational efficiency is not or rarely a problem to causal inference. On the contrary, we think that the absence of scaled causal inference is actually at the core of what is still missing for mordern machine learning systems. Imagine an unsupervised agent that can make all kind of inferences about the existing data and that can now make use of *a lot* of data. Now imagine a similar agent but that can do causal inferences. It is well established that causal inference is a superset to statistical inference (see [Peters et al. 2017](https://library.oapen.org/bitstream/handle/20.500.12657/26040/11283.pdf?sequence=1&isAllowed=y)) i.e., it basically just results in more issues than what we find in pure statistical inference. Furthermore, when you look at causal induction methods (one of the fundamental branches of causal inferences) - that is recovering the causal structure from the data alone - they scale very badly. For instance see NOTEARS by [Zheng et al. 2018](https://arxiv.org/abs/1909.13189) which estimates only linear SCMs, you can easily see that the method scales cubically with the dimensionality of the data (because of the matrix exponential in the acyclicity constraint $e^{W\circ W}$).
>
> * > $\color{purple}R:$ I think there is real novelty in a black box representation such as what the authors are presenting. The proposed framework is very general and could likely be of real utility for developing a wide range of downstream applications.
>
>   We certainly agree with you, as this is one of our key goals! Down the line we hope that there will be a tighter integration between everything that is useful in causality and all the best from modern machine learning. However, please don't forget that the key contributions that are most important to us are (1) that we try to engage in this research direction of integration between tractable models and causality which has been mostly negatively claimed by the results from [Papantonis and Belle (2020)](https://arxiv.org/abs/2001.10905) for which we give our answer (2) that tries to leverage the meta-operator $do(\cdot)$ introduced by Judea Pearl to be a differentiable function like a neural net, as we think this might push more into the direction of [causal-neural models (see Bareinboim et al. 2021)](https://arxiv.org/abs/2107.00793).

---

### Official Review · Reviewer_KcVJ · 2021-07-18

**Rating:** 2
**Confidence:** 3

**Summary:**

The paper proposes a new causal inference framework based on so-called ‘interventional Sum-Product Networks’ to target the problem of a lack of tractability in causal inference. It is based on standard SPNs and employs deep learning techniques to capture complex nonparametric functions describing multivariate conditional probability distributions. It is evaluated on several small synthetic data sets.


**Ethical Concerns:**

No ethical concerns

**Limitations And Societal Impact:**

No negative societal impact.

**Main Review:**

The problem is highly relevant, and the promise of having a means of reducing the computational complexity of causal (interventional) queries by obtaining sufficiently close but tractable approximations is very exciting.

However unfortunately the paper is extremely vague and implicit about the actual proposed methodology. After spending 5 pages describing known background, theory and other approaches, all in great clarity with some nice examples, the actual description of the contribution itself (p6) is surprisingly vague and without sufficient details to properly assess what is actually proposed or how to reproduce the results (see detailed comments below).
In fact, I am unsure whether the goal suggested at the beginning of the paper (‘here is a method that allows to compute interventional distributions from observational data + causal graph using an approximation based on tractable models’) is actually what the paper is about, or whether the main purpose of the paper is  to show that in principle SPNs could be used as tractable approximators of interventional distributions. If the first then I did not find it in the paper, if the second then it is hardly a surprising conclusion.

I am willing to entertain the notion I missed or misunderstood the main contribution of the paper (hence reduced confidence), but all things considered I doubt the paper is suitable for NeurIPS in its current form.

originality: novel suggestion, though unclear about the details
quality: poor: there is simply not enough detail to properly assess what is actually proposed and to what extent it makes sense or solves the initial problem statement
clarity: very clear on background, existing work and general examples (p1-5), but completely unclear when it gets to the actual contribution of the paper
significance: due to the lack of details the paper is unlikely to have significant impact in its current form

detailed comments:
p1.6 ‘subsuming’ suggests you claim to do it better
p1.23: ‘difficult to scale’ => why would they be? there are CBNs containing thousands to even millions of variables
p1.27: ‘weaving in the notion of interpretability’ => poetic but too vague
p2.36: ‘.. since a bipartite graph ..’ => bit sudden if you are not familiar with that ref.
p2.fig1: explain what the intervention ‘setting lung cancer to B(1/2)’ means
p2.43: ‘dream of tractable causal models is not insurmountable’ => yes but at what cost ? in general it has to be at best an approximation of the true interventional quantities right?
p2.57: ‘the inductive bias’ => perhaps introduce this first
p2.61-63: point 3. this is far too cryptic at this stage
p3.69: define CSPN before using acronym
p3.105: ‘causal mechanisms do not change through intervention’ => this is not what is meant by the assumption of ‘invariant causal mechanisms’: for example there are interventions corresponding to mechanism change, e.g. adding a catalyst to stimulate/enable a particular chemical reaction
p4.117: SCMs also allow non-recursive interactions (feedback); in addition not al variables need to be observed
p4.section 3.1: this insight is laudable but is standard common knowledge in the field of causal inference dating back at least to Reichenbach (1931) and before
p4.146: ‘curating’ is perhaps a bit much for generating data from such a tiny toy model …
p5,Fig2: ‘.. conditions on the mutilated causal graph ..’ => please explain better
also: figure is a tad small for such a crucial example => try to maximise (space permitting)

p5.173: so far we spent 5 pages without introducing anything new …
p6.176: move prefixed bold sentence ‘Definition of iSPN’ to line 184: now this paragraph is very confusing when you think you are finally getting to the ‘meat’ of the paper but then still end up just discussing other work first again

p6.185: ‘.. as inout the (mutilated) causal graph..’ => which one is it? the causal graph or the mutilated graph or both? (to be clear: you need both, or at least the original causal graph + target node of intervention)
idem: this definition (based on an adjacency matrix) does not seem to leave any room for possible unobserved confounders that were deemed so crucial to handle in p4.133?
p6.188, Def.1: this definition should be improved. at the moment there is nothing ‘interventional’ about this definition (even though you may intend to use it as such later).
also: ‘shared parameters’ => shared between what?

p6.191: being able to ‘answer’ a causal query is meaningless if you cannot provide guarantees about the outcome … and so far we have not seen anything that argues how or why it should be
p6.193: ‘The shared parameters allow for information flow during learning between the conditions and estimated densities’ => sounds great but I have no idea what it means as you have not explained any details yet

p6.294, Prop.1: ‘ .. with data D generated from the intervened SCM ..’ => ??? The problem is: we have data D from an unintervened system, and a corresponding causal graph … from this can we predict what will happen when we have intervene on Xi. If you assume you have data corresponding to this intervention the whole problem goes away. Or is this what you suggest to do in the paper: learn the structural equations from the data, set intervened equation to constant X_i = a, and sample new data from this modified system so you can then do standard inference on it given the intervened graph?

p6.217+: this is just so vague I have no idea what you are trying to say. Seriously, what does ‘freeing the investigation of iSPN from independent research around hidden confounding’ mean or why should we care?

p7.225: ok: here you are apparently learning the iSPN directly from the interventional data … if that is what the paper is about, then I’m afraid it has little new insights to offer … if not then please make sure to describe in detail exactly what you do.

p7-9: given that I still have no idea what you actually do or try to do I cannot assess the relevance or validity of the results presented here. This may be due to my lack of understanding (hence reduced confidence), but  tbh I don’t think that is the case here.

**Time Spent Reviewing:**

4

---

> ### Author Response · Authors · 2021-08-10
> **Author's response**
>
> Thank you for your review, $\color{purple}{\text{reviewer KcVJ}}$, we see that you went step-by-step into our work re-adjusting your insights along the way. At your last couple of 'details' comments (p7+), we observed a significant change in your understanding, being the reason you reduced your confidence as you felt you could not really asses our paper. We observed that you missed on some key points and we hope to clarify it for you now.
>
> * > $\color{purple}R:$ I am willing to entertain the notion I missed or misunderstood the main contribution of the paper (hence reduced confidence), but all things considered I doubt the paper is suitable for NeurIPS in its current form.
>
>   > $\color{purple}R:$ p7-9: given that I still have no idea what you actually do or try to do I cannot assess the relevance or validity of the results presented here. This may be due to my lack of understanding (hence reduced confidence), but tbh I don’t think that is the case here.
>
>   The following is a short summary of key aspects to our work that you completely missed in your review. Theoretically sound conversion schemes from SPNs to BNs exist, see [Zhao et al. 2015](https://arxiv.org/pdf/1501.01239.pdf). However, going the other way around from BNs to SPNs is already a completely different story because the classic way generally leads to a rather useless SPN (see Sec.4.3 of the same paper). But more importantly, just going from SPN to BN already leads to a degenerate BN which we point out to in our paper at two occasions (see p.1,l.32 and p.6,l.176). Therefore, it would be very surprising actually if SPNs could handle interventions! Furthermore, later in the history of this research direction, [Papantonis and Belle (2020)](https://arxiv.org/abs/2001.10905) give a strong ***negative*** claim on causality with SPNs/tractable models. However, they do *not* provide a negative proof and also suggest some possibilities where it could work, which they would not like to strike out. This is where our work now picks up, because with our key contribution and idea of leveraging the meta-operator $do(\cdot)$ introduced by Judea Pearl to be a differentiable function like a neural net. We think this can push in the right direction of resolving this very much long-standing issue. Our approach even has the benefit of pushing more into the direction of [causal-neural models (see Bareinboim et al. 2021)](https://arxiv.org/abs/2107.00793) to have a tighter integration even with modern deep learning!
>
> * > $\color{purple}R:$ However unfortunately the paper is extremely vague and implicit about the actual proposed methodology. After spending 5 pages describing known background, theory and other approaches, all in great clarity with some nice examples, the actual description of the contribution itself (p6) is surprisingly vague and without sufficient details to properly assess what is actually proposed
>
>   > $\color{purple}R:$ existing work and general examples (p1-5), but completely unclear when it gets to the actual contribution of the paper significance: due to the lack of details the paper
>
>   > $\color{purple}R:$ p5.173: so far we spent 5 pages without introducing anything new
>
>   > $\color{purple}R:$ when you think you are finally getting to the ‘meat’ of the paper but then still end up just discussing other work first again
>
>   This is the key theme of your review (simply based on the number of repetitions of this point throughout your review). This key theme of yours is about what you observed to be vague about our paper. Your statement suggests that we are wasting space, and we have to respectfully but strongly disagree with you. Furthermore, your (numerical) assessment is widely **exaggerated and not justified**. We will re-iterate. Depending on formatting about 35-55 lines compose a single page. We only spend 54 lines (meaning 1 page only!) (ref. l.67-121) on the complete background and related work section! For the preparation of our method, we only spend 49 lines (again 1 page only!) (ref. l.122-171). The Fig.2 naturally takes a lot of space but is carefully positioned to provide an intuitive overivew of the key technical concepts of the paper, for both more efficient communication of the ideas plus to in fact save space. Our main method (or 'meat') we discuss within 78 lines (approx. 2 pages!!) (ref. 172-250) which actually approaches the size of both previous sections together and is significantly larger than each of them! Also, aspects of the main method and how it relates to concepts like Pearl's Causal Hierarchy are already new insights being discussed early on, see p.1-2,l.29-52.
>
> * > $\color{purple}R:$ p4.section 3.1: this insight is laudable but is standard common knowledge in the field of causal inference dating back at least to Reichenbach (1931) and before
>
>   We strongly disagree with this view as the research at the intersection of causality and modern machine learning poses a significant minority within the whole community. Ideally, it should be standard but we strongly believe it isn't and treating it such way is inappropriate for any submission at modern venues. Looking at the statistics of popular venues (NeurIPS, ICLR, etc.) we can observe single digit percentages of causality-related papers are being submitted, and even less accepted because of exactly this issue of 'not being standard'.
>
> * > $\color{purple}R:$ The problem is highly relevant, and the promise of having a means of reducing the computational complexity of causal (interventional) queries by obtaining sufficiently close but tractable approximations is very exciting.
>
>   This is a point of strong motivation and relevance of our work we all agree on, this is exciting!
>
> * > $\color{purple}R:$ or how to reproduce the results (see detailed comments below).
>
>   We think our choices of presentation for reproducibility are adequate (the generality of the approach should not be confused with vagueness). Given that this points to a rather technical endeavour, it would be easiest to consider [our code that we made sure is reproducible over new setups](https://anonymous.4open.science/r/8a12a810-3343-4950-a5de-bd0721f4e914/README.md). It already provides even specific hyperparameters and the experiments we performed. To see how e.g. $g(\cdot)$ is being constructed or the neural network hyperparameters are set, please see Fig.2b and in the supplement p.2,l.58-64.
>
> **Our comments on the assessment of $\color{purple}{\text{reviewer KcVJ}}$**:
>
> * We appreciate the time the reviewer took for us. We have a few major issues with the review: (i) the review misses major aspects of our work which we tried to re-present in this rebuttal, (ii) the overly exaggerated negative impression on the spacing, (iii) bloating the review by repeating the negative impression of vagueness at least four times throughout, and finally (iv) not making explicit any positive points or strengths of our paper. We hope that we have clarified what the paper is about and that you are willing to reconsider your evaluation.

---

### Decision · Program_Chairs · 2021-09-28

**Decision:**

Accept (Poster)

**Comment:**

A few of the reviewers felt that the paper is in its current form too vague and would improve from a major revision.  For instance, they found the claim about how SPNs relate to causal inference not adequately developed. Unfortunately the rebuttal did not clarify the questions the reviewers raised. In light of this and the discussions and the reviews, I agree that the current version of the paper is not ready for publication.

**Consistency Experiment:**

NeurIPS has a long history of experimentation. In 2014, NeurIPS ran an experiment in which 10% of submissions were reviewed by two independent committees to quantify the randomness in the review process. This year, we repeated a variant of this experiment to see how the quality of the review process has changed over time.  This paper was part of the experiment and was therefore assigned to two committees (consisting of reviewers, an Area Chair, and a Senior Area Chair) that reached independent decisions.  If both committees made the same recommendation, this recommendation was followed. If a single committee recommended acceptance, the paper was accepted (with the exception of a few cases in which the other committee identified what we considered a fatal flaw, e.g., an error in a key result).

This copy’s committee reached the following decision: **Reject**

The other committee assigned to the paper recommended **Accept (Poster)**.  You can find the other set of reviews, along with any follow up discussion with the authors here:
https://openreview.net/forum?id=YMwraqG19Wg